# Automated Food Weight and Content Estimation Using Computer Vision and AI Algorithms

**DOI:** 10.3390/s24237660

**Published:** 2024-11-29

**Authors:** Bryan Gonzalez, Gonzalo Garcia, Sergio A. Velastin, Hamid GholamHosseini, Lino Tejeda, Gonzalo Farias

**Affiliations:** 1Escuela de Ingenieria Electrica, Pontificia Universidad Catolica de Valparaıso, Valparaíso 2340025, Chile; bryan.gonzalez.l@mail.pucv.cl (B.G.); gonzalo.farias@pucv.cl (G.F.); 2College of Engineering, Virginia Commonwealth University, Richmond, VA 23220, USA; garciaga3@vcu.edu; 3School of Electronic Engineering and Computer Science, Queen Mary University of London, London SE1 9RT, UK; 4Department of Computer Engineering, Universidad Carlos III de Madrid, 28911 Leganés, Spain; 5School of Engineering, Computer and Mathematical Sciences, Auckland University of Technology, Auckland 1010, New Zealand; hamid.gholamhosseini@aut.ac.nz; 6Sourcing, Chile; ltejeda@sourcing.cl

**Keywords:** food weight estimation, deep learning, computer vision, artificial intelligence

## Abstract

The work aims to leverage computer vision and artificial intelligence technologies to quantify key components in food distribution services. Specifically, it focuses on dish counting, content identification, and portion size estimation in a dining hall setting. An RGB camera is employed to capture the tray delivery process in a self-service restaurant, providing test images for plate counting and content identification algorithm comparison, using standard evaluation metrics. The approach utilized the YOLO architecture, a widely recognized deep learning model for object detection and computer vision. The model is trained on labeled image data, and its performance is assessed using a precision–recall curve at a confidence threshold of 0.5, achieving a mean average precision (mAP) of 0.873, indicating robust overall performance. The weight estimation procedure combines computer vision techniques to measure food volume using both RGB and depth cameras. Subsequently, density models specific to each food type are applied to estimate the detected food weight. The estimation model’s parameters are calibrated through experiments that generate volume-to-weight conversion tables for different food items. Validation of the system was conducted using rice and chicken, yielding error margins of 5.07% and 3.75%, respectively, demonstrating the feasibility and accuracy of the proposed method.

## 1. Introduction

The absence of reliable, quantitative data limits food distribution companies in making informed decisions and implementing continuous improvement strategies to optimize services. One of the key challenges addressed in this work is the application of technology to enhance food distribution and service processes. A critical gap identified is the lack of tools that can effectively monitor compliance with the conditions outlined in service contracts. This study aims to leverage computer vision (CV) and artificial intelligence (AI) technologies to systematically quantify the main variables commonly found in food service contracts. Specifically, the focus is on three key variables:Counting of the number of plates served over a specified time period.Identifying the contents of each plate by recognizing and individualizing each component of the dish.Measuring the portion sizes per plate, which involves estimating the weight of each component of the dish.

By quantifying these variables, it is expected that this system will enable control mechanisms such as ensuring compliance with the planned menu, measuring the quantity of food delivered, and monitoring the quality of dishes served. These controls are closely related to optimizing budget expenditure, enhancing process sustainability, and ensuring   consumer satisfaction.

Recent advancements in deep learning and computer vision have enabled the development of sophisticated food recognition systems that utilize image segmentation techniques for automatic dietary monitoring. For instance, ref. [1] explored the use of DeepLab-based deep convolutional neural networks (DCNNs) for food segmentation, focusing on distinguishing food and non-food regions, a critical step in identifying individual dish components in complex meal settings. Their results, applied in controlled environments like cafeterias, highlight the importance of accurate segmentation in subsequent stages like food quantity estimation, a task directly relevant to the portion size measurement targeted in this work. Similarly, ref. [2] introduced a segmentation and classification system tailored to Brazilian food items. Their approach, which compares state-of-the-art segmentation models such as Mask R-CNN and FCN, shows the effectiveness of these models in accurately recognizing and segmenting food in diverse settings. This method provides valuable insights for automating content identification in multi-component dishes, a challenge we aim to address in the identification of dish components. More recently, ref. [3] proposed the FoodSAM framework, an extension of the segment anything model (SAM), which incorporates zero-shot capabilities, for instance, panoptic, and promptable segmentation. This system demonstrates significant advancements in food image segmentation by recognizing individual food items and related objects, such as cutlery and plates, contributing to a more granular understanding of food composition. The inclusion of instance segmentation in this work parallels our need to individualize dish components for more accurate portion size estimation.

The proposed solution automates the quantification of plate count, content identification, and portion measurement in a dining hall setting. An RGB camera is used to capture video footage of the tray delivery process in a restaurant, which is then utilized to generate a dataset of training images. These images are processed by state-of-the-art AI algorithms, which are then evaluated using standard performance metrics. The YOLO model ref. [4], a well-established deep learning network (DLN) for object detection, is employed for this task. The model is trained on labeled images representing different classes, aiming at accurate detection and classification in subsequent analyses. This approach aligns with studies that have demonstrated the effectiveness of DLNs in recognizing diverse food items based on their shapes, sizes, and textures [5,6,7,8,9,10,11,12].

For example, ref. [13] developed a deep-learning-based method for calorie estimation using computer vision techniques to detect and segment food items, estimating their volume and corresponding caloric content. Similarly, ref. [14] introduced a system that utilizes depth cameras to capture the volume of food and accurately estimate calories by combining RGB and depth information. While calorie estimation is not part of the current study, these approaches demonstrate how food recognition systems could be extended to include nutritional analyses, providing valuable insights for dietary assessments.

The proposed methodology for portion weight estimation in this work also relies on a combination of RGB and depth cameras, similar to ref. [15], who developed a system integrating segmentation and weight estimation based on image analysis. The work   in ref. [16] took a different approach by using cutlery as reference objects for weight estimation, achieving reliable results with minimal user input. Other approaches, such as using cutlery as reference objects for weight estimation, have also achieved reliable results with minimal user input [16].

The volume is extracted by a numerical process derived from the depths of each image pixel after the automatic segmentation of the ingredients and previous conversion to millimeter units. The depth of the dish and the tray are important parameters to be considered during each set of measurements. The final weight per ingredient is calculated by applying their density from a look-up array obtained in the previous step of dish ingredient identification. In this study, the weight estimation procedure relies on CV devices and techniques, integrating data from an RGB camera and a depth camera to measure the volume of each food component. The Luxonis OAK-D Lite camera [17], which combines an RGB color camera with stereo depth cameras in a single device, is used to capture food volume.

This approach depends on the existence of detectable texture on the food’s surface. The employed passive depth measurement in this work is carried out on typical food served in cafeteria, restaurants, or other self-service places. Nowadays, stereo-based cameras are cheaper than other technologies for this application, so they were selected. In case the food has low-texture, active measurement such as laser or IR pattern can be applied. There are many alternatives to stereo-based cameras (ToF, LiDAR, etc). Another important consideration in this work is the assumption that the plates (and some other associated conditions) should be very regular, as the proposed solution was designed for industrial food service. However, in case conditions change, we could adapt or retrain the model to a particular scenario (plate shape, tray design, type of food, etc.).

Density models specific to each food type are then applied to estimate the weight based on volume measurements. Calibration of the estimation model’s parameters is conducted through experiments, generating volume-to-weight conversion tables for different food items, allowing for accurate weight estimation tailored to each food type.

The paper’s main contributions are as follows: Ingredient Identification: Using computer vision algorithms and deep learning networks as applied to RGB camera images to identify ingredients in dishes. These findings are then used to retrieve mean density values from a look-up array, aiding in weight estimation.Volume Determination: Automated volume calculation of dish ingredients using a sequence of RGB-depth images. We segmented RGB images of the dish ingredients, dish, and tray and used pixel depth measurements from the depth data, converting it to millimeters for geometric volume calculation.Automated Weight Estimation: Full automation and integration of the identification and volume calculation steps to deliver accurate weight estimations for food ingredients.

This work is structured into two parts: the first part describes the development of a system for plate counting and content identification using CV and AI tools, while the second part focuses on techniques for measuring the weight of each food component. Detailed descriptions of both hardware and software configurations are provided in each section.

## 2. Materials and Methods

### 2.1. Camera Setup

The selected camera is the Luxonis OAK-D Lite model [17] (Westminister, CO, USA), known for its ability to combine RGB and depth (RGB-D) images in a single device. This camera uses a stereo system with two coplanar lenses separated by a fixed distance, which allows depth to be calculated using the disparity between the two captured images to obtain three-dimensional food data. Video capture was set at a rate of 30 frames per second (FPS), using the H.265 compression format, which allows efficient data storage and transmission without compromising image quality. The camera was installed at a strategic point above the serving line in a self-service restaurant, ensuring a clear and unobstructed view of the trays with the main dishes, which are the objects of interest in this study. To ensure the accuracy of the depth measurements, the following adjustments were made to the camera settings (see Table 1):

As depicted in Figure 1, the system interconnection consists of the RGB-D camera [17], comprising the RGB camera and two stereo cameras, mounted on a support located at a height of approximately 1.2 m above the food tray. The RGB camera captures the color image of the object, while the stereo cameras generate a depth map of the scene.

For the training of the food detection model, a task that is performed off-line before operation, the Hikvision model DS-2CD2743G2-IZS camera [18] (Hangzhou, Zhejiang, China) was used. This device was selected for its ability to provide high-resolution images and its adaptability to different lighting conditions. The camera has a maximum resolution of 2688 × 1520 pixels and is equipped with a 2.8 to 12 mm varifocal lens, which allows the field of view (FOV) to be adjusted according to the experiment condition. The horizontal FOV varies from 95.8° to 29.2°, while the vertical FOV ranges from 50.6° to 16.4°.

### 2.2. Methodology

Figure 2 shows the flowchart of the image processing pipeline used in the study. The initial setup involves the capture of both RGB and depth frames. Once the frames are captured, the process checks for the presence of food. If food is detected, the pipeline proceeds with the segmentation process, followed by volume estimation and finally, the estimation of the food weight.

### 2.3. Initial Setup

The initial setup involves preparing and configuring the camera parameters for capturing both RGB and depth images. This includes defining the resolution, applying filters, and other specific adjustments, as described earlier. Before proceeding with capturing images of the food, an initial check is conducted by capturing an image of a flat surface. This ensures that the camera is accurately measuring distance, verifying that the configuration parameters are appropriate for precise data capture during the experiment.

### 2.4. RGB Frame Capture

RGB frame capture is performed using the RGB camera integrated into the OAK-D Lite, which also includes two stereoscopic cameras for depth capture. No filters or color adjustments are applied to the RGB images. However, a downscaling process is conducted to synchronize the resolution of the RGB images with the depth images, as the cameras have different native resolutions. The capture operates at a frequency of 30 FPS, ensuring that sufficient data are obtained for detailed and accurate analysis in subsequent stages. An example of the captured RGB image can be seen in Figure 3.

### 2.5. Depth Frame Capture

Depth frames are captured using the stereoscopic cameras of the OAK-D Lite. The depth maps are generated using the disparity technique, with depth calculated using the following formula:(1)$$\mathrm{depth}=\frac{\mathrm{focal}\;\mathrm{length}\;\mathrm{in}\;\mathrm{pixels}\cdot \mathrm{baseline}}{\mathrm{disparity}\;\mathrm{in}\;\mathrm{pixels}}$$
where

focal_length_in_pixels is the focal length of the camera expressed in pixels. This value is obtained by multiplying the physical focal length (in millimeters) by the camera’s conversion factor (pixels per millimeter).baseline is the physical distance between the two camera sensors (in millimeters) in a stereo system.disparity_in_pixels is the difference in pixels between the positions of an object in the two images captured by the stereo cameras.

To enhance the accuracy of depth measurements, specific parameters as described in the initial setup are applied, and post-processing filters, specifically the temporal filter and spatial filter, are used to optimize the quality of the depth images before they are utilized in subsequent analysis processes.

An example of the captured RGB and depth image can be seen in Figure 4a,b.

### 2.6. Food Detection

The food detection process begins with the video sequences recorded using the RGB camera. The first step involves segmenting these videos into individual frames, which are then labeled to prepare for training the detection model.

#### 2.6.1. Dataset and Training Configuration

For the training process, the dataset consisted of 11,424 training images, 1428 test images, and 1428 validation images, where the division was made randomly. This division ensured a broader spectrum of examples, allowing the model to learn from a wide variety of scenarios and object variations. The diversity and number of images is aimed at improving the model’s adaptability and generalization ability.

The images included a diverse range of food items commonly found in the dining hall, such as trays, pork, sausages, noodles, chickpeas, and various types of plates. The dataset was carefully curated to ensure a balanced representation of each class, which is crucial for reducing bias and enhancing the model’s performance across different categories. All the classes that were used for training are shown in Figure 5.

#### 2.6.2. Training Duration and Computational Resources

The training duration for this model was approximately 14 h on a system equipped with an NVIDIA RTX 2060 GPU [19] (Santa Clara, CA, USA). This training time reflects the complexity of the model and the extensive dataset used. Future training sessions could benefit from parallel processing on multiple GPUs, which would reduce the training time and potentially improve the model’s performance by allowing for more epochs and finer hyperpara-meter tuning.

### 2.7. Segmentation Process

The segmentation of food items within the bounding box is performed using a fine-tuned YOLOv8L-Seg model [20]. This process is fully automatic and was trained on a dataset of 1025 images. Unlike traditional bounding box methods, this segmentation is carried out at the pixel level, providing precise outlines of the food items.

#### 2.7.1. Dataset Annotation

The dataset used for training the segmentation model consisted of 1025 images, annotated manually to ensure high-quality segmentation masks. The annotation process was carried out using the Roboflow platform, which provides an intuitive interface for pixel-level annotation. A total of 14 classes were defined for segmentation, representing the most common food items and objects in the dining hall setting. Each image in the dataset was carefully annotated by manual tracing of the object contours to create precise segmentation masks. This manual annotation ensured that the model could learn from accurate examples, improving its ability to generalize to new images.

An example of an annotated image from the dataset is shown in Figure 6.

#### 2.7.2. Training and Evaluation Metrics

To evaluate the performance of the segmentation model, we trained the YOLOv8L-Seg model for 200 epochs using the Ultralytics framework. The training configuration included an image size of 640×640 pixels, a batch size of 32, and automatic mixed precision (AMP) to optimize computational efficiency.

To further analyze the model’s performance, we obtained a precision–recall (PR) curve (see Figure 7) that illustrates the relationship between precision and recall across different confidence thresholds.

The mAP@0.5 across all classes reached 0.895, indicating that the model performs robustly in segmenting most food items. High-performing classes, such as ‘Fish’, ‘Hamburger’, ‘Meat’, and ‘Tomato’, achieved mAP values close to 1.0, reflecting excellent segmentation precision and recall. These results suggest that the model can effectively outline these items with minimal false positives and false negatives, maintaining high reliability in varied conditions.

However, the model’s performance is not uniform across all classes. For instance, the class ‘Tray’ exhibited a notably low mAP of 0.404, signaling difficulties in accurately segmenting this item. This confusion may be due to the tray being surrounded by many elements such as plates, spoons, forks, knives, and desserts, which reduces the number of pixels available for labeling and suggests that a larger number of images may be required to improve this reliability parameter.

An example of this automatic segmentation can be seen in Figure 8, where rice, chicken, and french fries were correctly detected, but the tray was not detected.

### 2.8. Volume Estimation

The estimation of object volumes from captured images is a critical component of this study, necessitating precise calibration to convert pixel measurements into millimeters. The procedure begins with the determination of a specific conversion factor, which is dynamically adjusted according to the distance between the camera and the object  of interest.

This conversion factor is applied to the width and height in pixels of each identified volumetric element, often referred to as a “bar”. By applying this factor, the dimensions of each bar are converted from pixels to cubic millimeters, allowing for an accurate estimation of the object’s volume. The total volume is calculated by summing the volumes of all bars, as expressed in the following equation:(2)$$V=\sum \left(\sup\cdot k_{\mathrm{pix}}^{2}\right),\;\left[{\mathrm{mm}}^{3}\right]$$

For enhanced visualization and interpretability, a three-dimensional mesh is generated from the volume data, using the millimeter-converted values. This mesh is constructed based on a grid that is scaled according to the pixel conversion factor $k_{\mathrm{pix}}$, aligning with both the u- and v-axes of the image dimensions. The axes are labeled in millimeters to facilitate an accurate understanding of the physical dimensions of the analyzed object. The final 3D visualization, typically adjusted for a top view, provides a clear and intuitive representation of the volumetric distribution across the object.

### 2.9. Automatic Segmentation with Model

To streamline the volume estimation process, an automatic segmentation model is used, which requires minimal manual intervention as opposed to what is seen in [21].

Fixed Reference Point: Unlike manual segmentation, a fixed reference point is used in the automatic process. This point is pre-determined and remains constant across all images, ensuring consistent volume measurements.Application of Automatic Segmentation Model: The automatic segmentation model detects and segments the object’s boundaries. The model, trained on a variety of image data, applies pattern recognition to accurately identify and segment the object within the image.Volume Estimation: The segmented contour obtained from the automatic model is used to calculate the volume. With the reference point fixed, the pixel measurements are converted to millimeters, and the total volume is computed following the stan-  dard procedure.

Figure 9 shows the volume estimation for the same dish with rice, chicken, and   french fries.

### 2.10. Volume Calculation

The volume calculation process is summarized in Algorithm 1, which outlines the key steps of calibration, segmentation, and volume estimation:
**Algorithm 1** Calibration, Segmentation using YOLOv8L-Seg, and Volume Calculation
 1: Calibration:
 2: Load depth and RGB image data
 3: Ensure correspondence in terms of pixel size and orientation between RGB and depth images in case the RGB and depth images come from different cameras
 4: **Segmentation:**
 5: Apply YOLOv8L-Seg to the RGB image to obtain segmentation masks
 6: Extract the mask corresponding to the object of interest
 7: Create a binary mask in(s,k)∈0,1 indicating object pixels of the RGB image, which has dimensions of *s* and *k*
 8: **Depth Data Processing:**
 9: Apply depth saturation: $depth_{\mathrm{sat}}=\max\left(\min\left(depth,\;sat_{\max}\right),\;sat_{\min}\right)$
10: Set the reference depth value RP from a known reference point in the tray
11: Convert pixel measurements to millimeters using $k_{\mathrm{pix}}$
12: **Volume Calculation:**
13: Compute the surface contribution: $sup\left(s,\;k\right)=in\left(s,\;k\right)\times \left(RP-depth_{\mathrm{sat}}\left(s,\;k\right)\right)$
14: Calculate total volume using Equation (2)

Where

depth: The original depth map obtained from the depth sensor, representing the distance from the camera to each point in the scene.$depth_{\mathrm{sat}}$: The saturated depth map after applying minimum and maximum thresholds, defined by $sat_{\min}$ and $sat_{\max}$.RP: Depth value at the reference point. This value serves as a baseline to measure relative depth differences in the scene.in(s,k): Pixels inside the segmentation mask border.sup(s,k): Surface contribution at the pixel position (s,k), calculated as
$$sup\left(s,\;k\right)=in\left(s,\;k\right)\times \left(depth_{\mathrm{sat}}\left(s,\;k\right)-RP\right)$$

It represents the height or elevation of the object at each pixel relative to the reference point. The total volume is computed by Equation (2).

### 2.11. Food Weight Estimation

Following the segmentation and volumetric quantification of the food items, the calculated volume becomes a critical input for the weight estimation model. The primary goal of this process is to develop a machine learning model capable of accurately predicting the weight of the food based on its volume, which is particularly relevant for applications in food management and quality control.

The volumetric data obtained from the previous steps serve as distinguishing features for training the predictive model. The model is designed to learn the complex relationship between the volume of the food item and its corresponding weight. This relationship is often non-linear, requiring a sophisticated modeling approach to capture the subtleties inherent in the data.

The volume of proteins, such as chicken, is obtained using depth images captured by the depth camera. By identifying and segmenting the protein in the images, we use the YOLO models that were previously mentioned. This volume is measured in cubic millimeters and is then used, as a feature, to estimate its weight.

To train a model capable of estimating the weight of segmented food items on a plate, it is essential to obtain accurate measurements for these foods. For this purpose, an electronic scale was used to weigh each food item individually.

To develop a reliable model, multiple machine learning algorithms were evaluated, with the exponential Gaussian process regression (Exponential GPR) model [22] emerging as the most effective. This model is particularly well-suited for scenarios where the relationship between the features (in this case, the volume) and the target variable (weight) exhibits smooth, continuous variations. The Exponential GPR model uses an exponential kernel, also known as the Ornstein–Uhlenbeck kernel, which is adept at handling scenarios where the impact of observed features decays exponentially over time or distance.

The exponential kernel is mathematically defined as
(3)$$k\left(x,x^{\prime }\right)=\sigma_{f}^{2}\exp\left(-\frac{∥x-x^{\prime }∥}{2l^{2}}\right)$$
where

$\sigma_{f}^{2}$ represents the signal variance, determining the amplitude of the covariance.*l* is the length scale parameter, defining the rate at which correlations diminish with increasing distance between data points.

Although the input to our model is the estimated volume of the food, it is important to note that the volume estimation process is inherently influenced by the surface texture of the food. The depth camera we use employs a stereoscopic principle to generate depth maps, which is based on identifying matching features between two slightly offset images. On smooth or insufficiently textured surfaces, depth estimation can be inaccurate because the camera struggles to find the corresponding points between the images. This inaccuracy in depth perception causes errors in the calculated volume. Consequently, texture variations indirectly affect the volume measurements provided to the model. The exponential Gaussian regression model (exponential GPR) is particularly suitable in this context because it can handle the small variations and uncertainties in the input data caused by these texture-induced inaccuracies. By effectively capturing dependencies and continuity in the data, the model improves weight estimation despite the challenges posed by texture variations.

In practice, the weight estimation process is validated through cross-validation techniques with 5 folds, so that the model generalizes well to unseen data. The accuracy of the model is further enhanced by comparing the predicted weights with the actual measured weights, allowing for continuous refinement of the model.

For a comprehensive understanding of the model’s behavior, the distribution of prediction errors is analyzed. This analysis provides insights into the model’s accuracy and highlights any potential biases or areas for improvement. Although outliers may occur, these instances offer valuable opportunities to refine the model and improve its robustness across a wide range of conditions.

The overall goal of this weight estimation process is to provide a reliable and accurate tool for assessing the weight of food items, contributing to more efficient and effective food management practices.

## 3. Experimental Results

### 3.1. Dining Hall

The detection is performed on the RGB image, where the YOLOv8 model identifies the presence of food items by generating bounding boxes around them. This bounding box is crucial, as it will later be used in conjunction with the depth image to calculate the volume of the food, a process explained in subsequent sections.

To obtain an accurate detection, a confidence threshold of 50% is applied. If the confidence level for a detected item falls below this threshold, the image is discarded. This is because insufficient confidence would undermine the segmentation process and the subsequent volume calculation, leading to inaccurate results.

A successful example of food detection is shown in Figure 10

#### 3.1.1. Confusion Matrix Analysis

The confusion matrix provides insight into the model’s prediction accuracy for each class and highlights areas where the model confuses different classes. For example, the class “Tray” achieved a high accuracy with a value of 0.93 on the diagonal of the confusion matrix, indicating that 93% of the time, the model correctly predicted the “Tray” class. However, some classes showed notable confusion, suggesting that additional data or more precise labeling could improve the model’s performance in these areas.

The confusion matrix obtained is shown in Figure 11.

#### 3.1.2. Precision–Recall Curve and Error Analysis

The precision–recall (PR) curve further illustrates the model’s performance across different thresholds. Ideally, the PR curve should approach the top-right corner, indicating high precision and recall. The average precision across all classes remained relatively high for most levels of recall, demonstrating the model’s ability to maintain a balance between precision and recall even at higher thresholds. The mAP value of 0.863 confirms that the model performs well across various classes, although there is a noticeable drop in precision as recall approaches 1.0, which is typical as it becomes more challenging to maintain high precision with very high recall.

Error analysis revealed specific areas for improvement. Certain classes exhibited significant confusion, indicating the potential benefit of refining the training data or increasing the number of training examples for these specific categories. The precision–recall (PR) curve is illustrated in Figure 12.

### 3.2. Depth Camera

The validation process for the weight estimation model in rotisserie chicken images was conducted using an expanded dataset comprising 72 different images. These 72 images correspond to five different dishes that are presented in a rotating manner. In each image, the position of the elements on the tray is varied, so that the side dish, protein, and main plate change places. After applying the established methodology, the corresponding training was performed, resulting in ten models with remarkable accuracy, whose root mean square error (RMSE) metrics are presented in Table 2. Notably, the exponential Gaussian regression (Exponential GPR) model exhibited the lowest RMSE, highlighting its superior predictive accuracy. To verify the reliability of the selected model, the percentage error in the predictions was calculated through a cross-validation procedure applied to the 72 images. The results showed an average error of only 2.86%, demonstrating the model’s high accuracy in estimating weights within the evaluated context.

It is a common assumption that the weight estimation of food items based on their volume should follow a linear relationship. However, our empirical observations indicate that this is not the case for the food items analyzed. The exponential Gaussian process regression (Exponential GPR) model, which captured non-linear dependencies, achieved the lowest RMSE among the models tested.

These reveal that the relationship between volume and weight is indeed complex and varies across different food items. Factors such as varying densities, irregular shapes, cooking time, supplier variability, and preparation methods, and heterogeneous compositions contribute to this complexity, making a simple linear model insufficient for accurate weight estimation.

Back to the best model obtained, the histogram shown in Figure 13 depicts the distribution of errors in the weight predictions, providing quantitative insight into the model’s accuracy. The highest frequency density is observed around the 0 to 10 error range, indicating that most predictions have minimal error. This concentration near the origin suggests high accuracy in weight estimation, with a balanced distribution of positive and negative errors, reflecting the absence of systematic bias towards over- or underestimation by the model.

However, outliers with significant errors were identified, indicated by the lower frequencies at the extremes of the error range. These instances highlight the need for further evaluation to determine the underlying causes, which may range from limitations in the data capture device used to inherent limitations in the prediction algorithm.

In this case, a detailed analysis of one of the most relevant outliers is conducted. Table 3 shows a comparative analysis of the volume of the chicken legs.

Where

Comp. (Complete Plate): Refers to the dish with its full initial accompaniment.1/2 Acc. (Half Accompaniment): The accompaniment is reduced to half of the initial amount.1/4 Acc. (Quarter Accompaniment): The accompaniment is reduced to a quarter of the initial amount.1/8 Acc. (Eighth Accompaniment): The accompaniment is reduced to one-eighth of the initial amount.

Here, we observe an initial indication that the model could generate predictions with a degree of error. Specifically, the estimated volumes in full plate configuration 2 (highlighted in yellow) are considerably lower compared to those estimated in other configurations. It should be noted that the same protein was used in all measurements, so minimal variations between estimated volumes would be expected. However, these discrepancies may be subject to human error during the manual segmentation process.

To gain a more detailed understanding of the estimated volume, we proceed to analyze the depth image of these cases of interest. First, we begin by studying the depth image of configuration one (C1, Comp.) of the full plate. Below, Figure 14 illustrates the depth image for this first case study.

As shown, it is evident that the camera is not detecting the protein correctly. In particular, the top of the chicken appears flattened. This could be due to two main factors: first, the stereo camera may be reaching the limits of its height capability; second, the absence of sufficient texture on the top of the chicken may prevent the depth camera from adequately capturing the details necessary for accurate detection.

Additionally, the detection fails to adequately represent the shape of the protein, especially in areas where chicken bones protrude, which are not correctly identified by the camera. An example of successful protein detection, where the structural features are clearly defined, is illustrated in Figure 15.

It is important to note that Figure 15 was obtained using configuration 2 when only an eighth of the accompaniment remained on the plate. The correct detection of the protein in this configuration was expected, as Table 3 also reveals an interesting trend in the standard deviation of the estimated volumes for each configuration. It is notable that this parameter progressively decreases as the protein becomes more isolated, indicating greater consistency in the measurements as the presence of the accompaniment is reduced.

### 3.3. Weight Estimation for Rice

A procedure similar to that used for rotisserie chicken was applied to estimate the weight of rice portions. In this case, it was observed that the relationship between the estimated volume and the actual weight of the rice follows a linear trend, confirming the initial assumption that weight estimation based on volume can be modeled linearly for certain types of food.

#### 3.3.1. Dataset and Methodology

A dataset of 48 images of rice portions was collected, with each portion individually weighed using an electronic scale to obtain labels. The segmentation of the rice was performed using the YOLOv8L-Seg model, effectively isolating the rice in the images. The volume of each rice portion was estimated from the depth images, following the methodology described in Section 2.6 The volumetric data served as input features for training machine learning models to predict the weight of the portions.

#### 3.3.2. Models Evaluated and Results

Several machine learning models were evaluated, including linear support vector machine (Linear SVM), exponential Gaussian process regression (Exponential GPR), rational quadratic GPR, and others. The RMSE values for the top-performing models are presented in Table 4.

The Linear SVM model achieved the lowest RMSE of 30.809 grams, indicating that a linear model is adequate for estimating the weight of rice based on its volume. This suggests that the relationship between volume and weight for rice is effectively linear. Figure 16 shows the predictions of the model when using cross-validation with 5-fold.

## 4. Conclusions and Future Works

The work laid the foundation for a food distribution monitoring system within a casino setting. Through real-world experiments, procedures for food ingredient identification and portion size estimation were developed and validated. Traditional RGB cameras, integrated with YOLO neural networks, proved effective for food identification, provided that high-quality training images are used. Portion weight estimation was achieved by measuring food volume on plates using a specialized procedure that integrates RGB and depth cameras. This procedure involves training an estimation model tailored to specific food types, such as rice and chicken, with estimation errors of 5.07% and 3.75% respectively. Stereo cameras were also validated for their precision in distance measurement, particularly when equipped with complementary lighting patterns.

Future work will focus on implementing the solution in Python 3.11 or C++, evaluating cloud versus local server deployment, and developing a user-friendly reporting system to facilitate practical management. Stereo cameras were used to measure food depth, relying on its surface texture. If texture is low, laser or IR can be an alternative due to affordability. LiDAR technology, for example, enhances depth measurement accuracy, especially on smooth surfaces where other cameras face challenges. Using structured light projection can improve depth measurement precision for irregular food shapes, offering a better solution than current methods. On the other hand, a data imbalance led to potential overfitting issues, especially evident in poor potato classification results. To address this, augmentation techniques are recommended for balancing data and overcoming these challenges. Furthermore, thorough cross-validation with diverse datasets is crucial to assess the model’s generalization to new data. Moreover, these algorithms could be used to handle the scenario by extracting data from visible trays, discarding incomplete ones, and tracking multiple trays in real time. Finally, it will be interesting to address in the future when occlusion occurs between two different types of food.

## Figures and Tables

**Figure 1 sensors-24-07660-f001:**
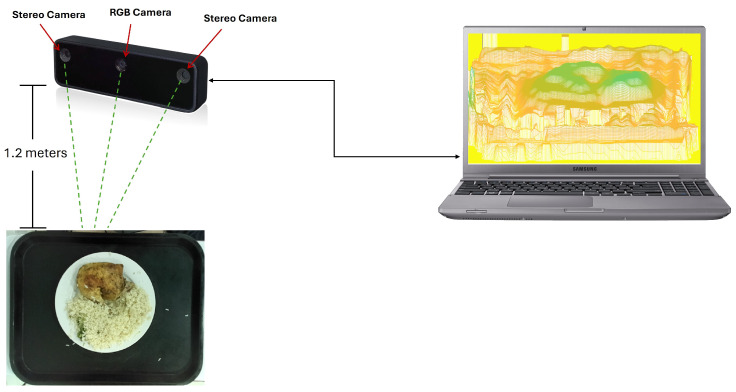
Interconnection between OAK-D lite camera and computer and distance to the tray.

**Figure 2 sensors-24-07660-f002:**
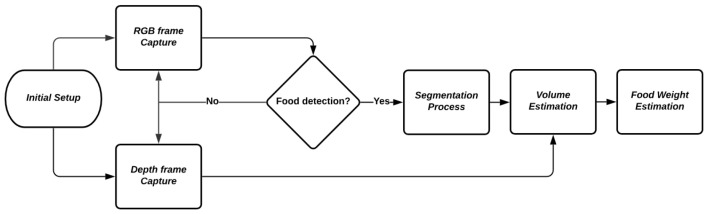
Flowchart of the image processing pipeline used in the study.

**Figure 3 sensors-24-07660-f003:**
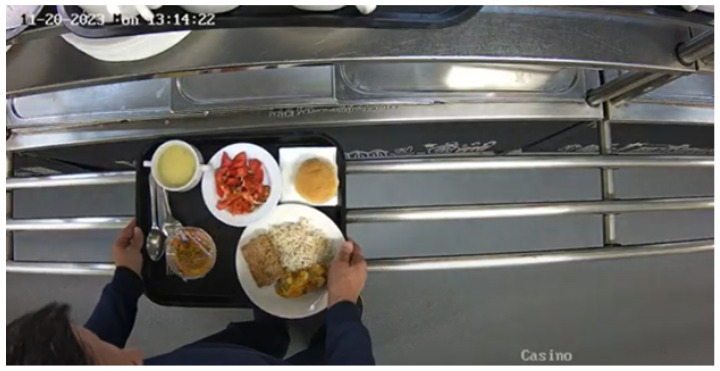
The image was taken from a top-down view in a cafeteria-like setting, with a timestamp displayed in the top left corner.

**Figure 4 sensors-24-07660-f004:**
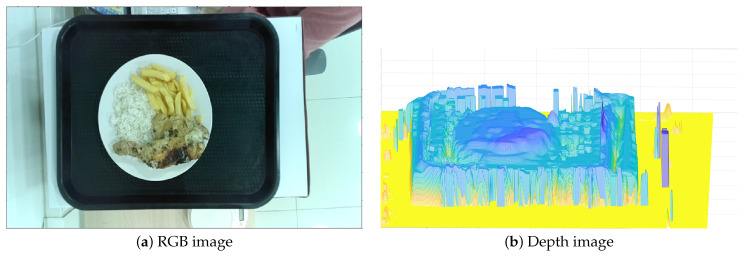
RGB and depth image captured chicken leg dish with rice and french fries.

**Figure 5 sensors-24-07660-f005:**
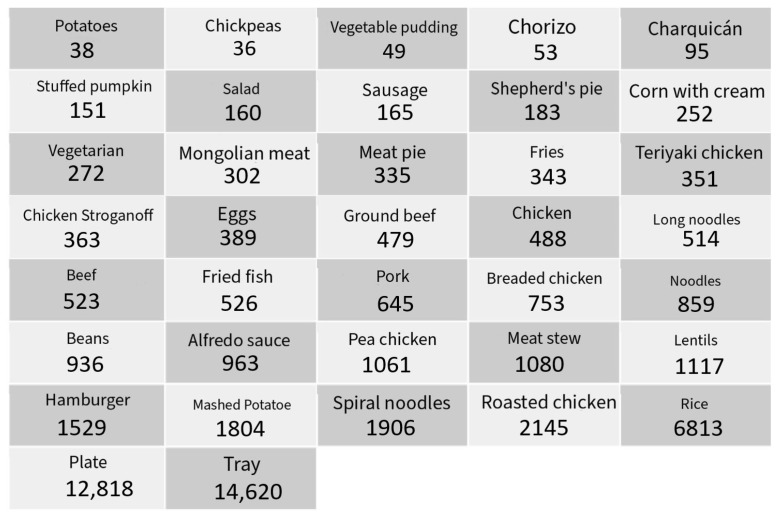
All classes used for detection training.

**Figure 6 sensors-24-07660-f006:**
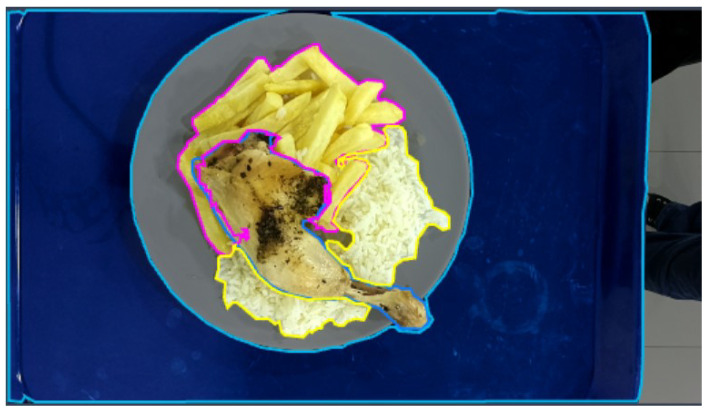
Example of an annotated image from the dataset, showing the segmentation masks for different food items (light blue is for tray, dish, and chicken, yellow is for rice, magenta is for French fries).

**Figure 7 sensors-24-07660-f007:**
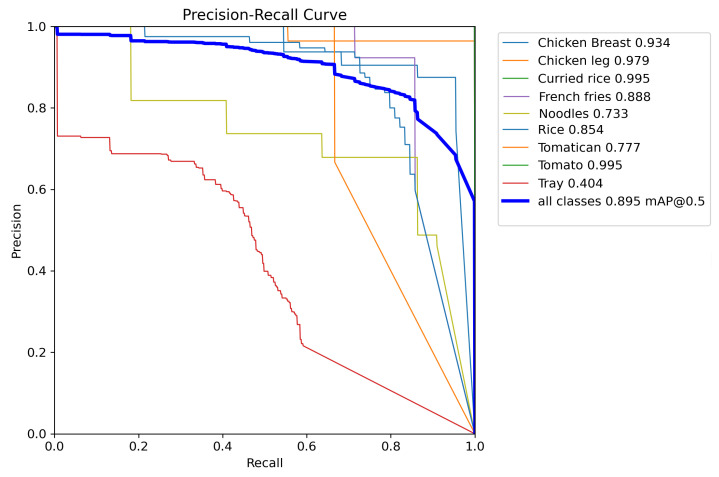
Precision–recall curve for the YOLOv8L-Seg model on the segmentation task in testing data. The curve shows the trade-off between precision and recall at different confidence thresholds.

**Figure 8 sensors-24-07660-f008:**
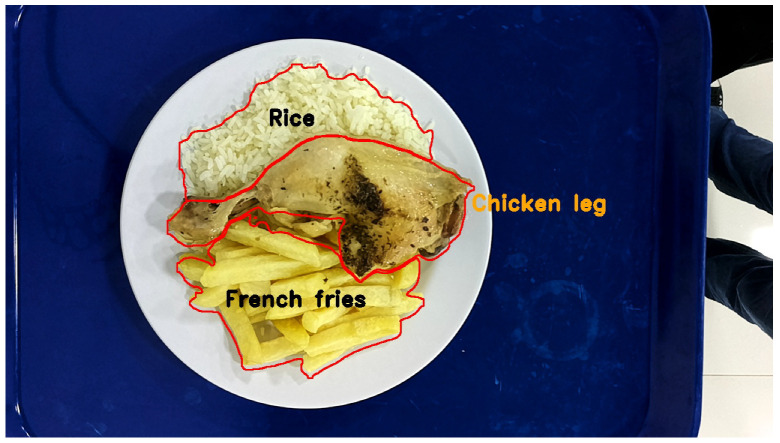
Automatic segmentation process using YOLOv8L-Seg.

**Figure 9 sensors-24-07660-f009:**
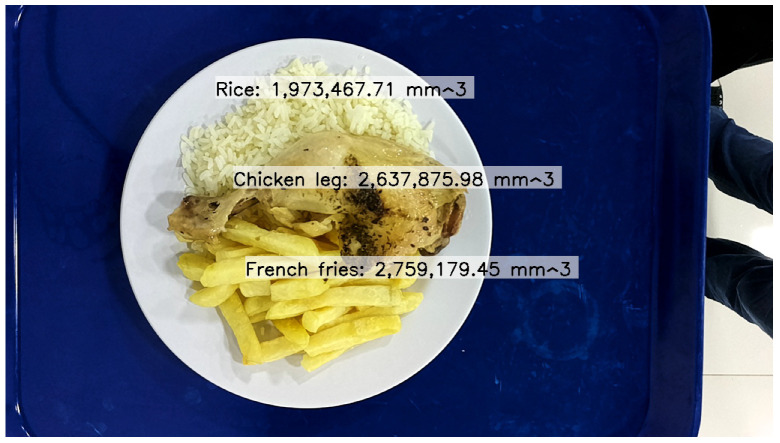
Automatic segmentation and volume estimation process illustrating the fixed reference point and automatically detected contours of the object.

**Figure 10 sensors-24-07660-f010:**
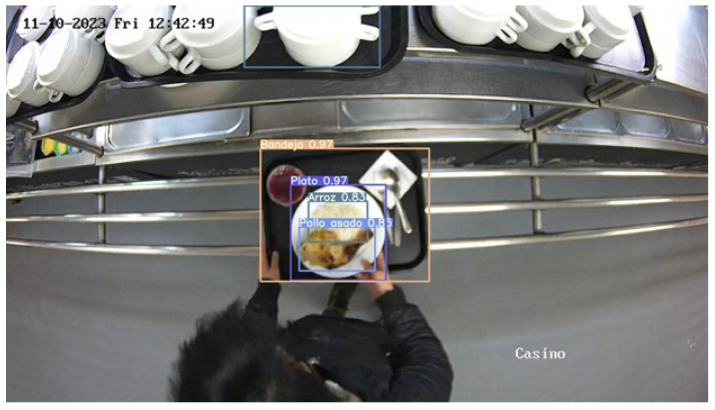
The system has successfully identified and labeled various elements on the tray, including the tray itself, a plate, roasted chicken, and rice, each with corresponding confidence levels. The image also includes a timestamp and is set in a cafeteria environment.

**Figure 11 sensors-24-07660-f011:**
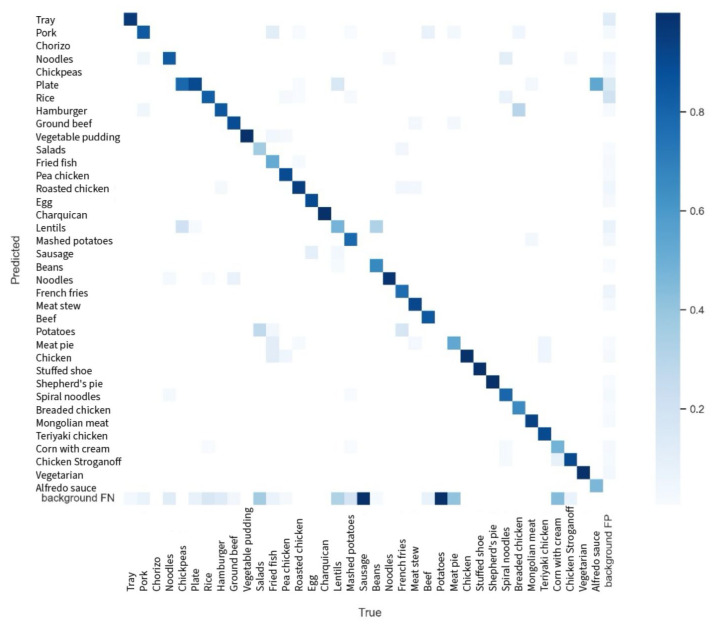
Confusion matrix using the testing data.

**Figure 12 sensors-24-07660-f012:**
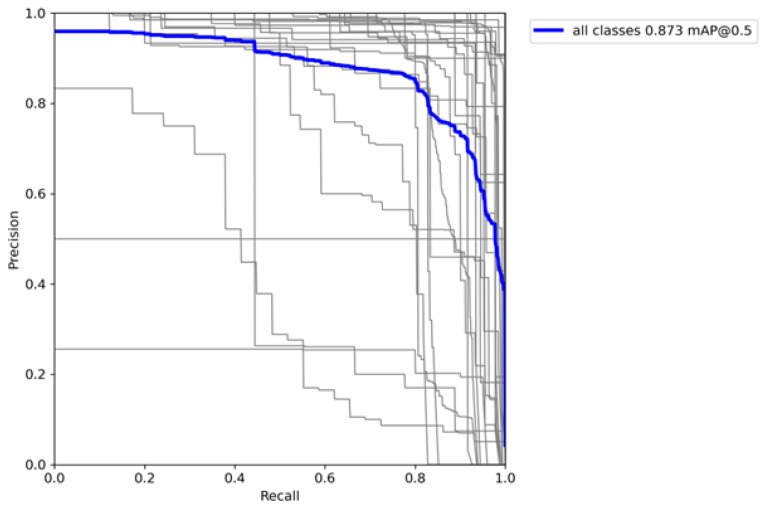
Precision–recall curve. The gray lines represent the individual precision-recall curves for each class evaluated in the model, showing variability in performance across classes. The blue line corresponds to the weighted average precision-recall curve across all classes, indicating the overall performance of the model. The value of 0.873 mAP@0.5 represents the mean average precision (mAP) across all classes at an intersection-over-union (IoU) threshold of 0.5.

**Figure 13 sensors-24-07660-f013:**
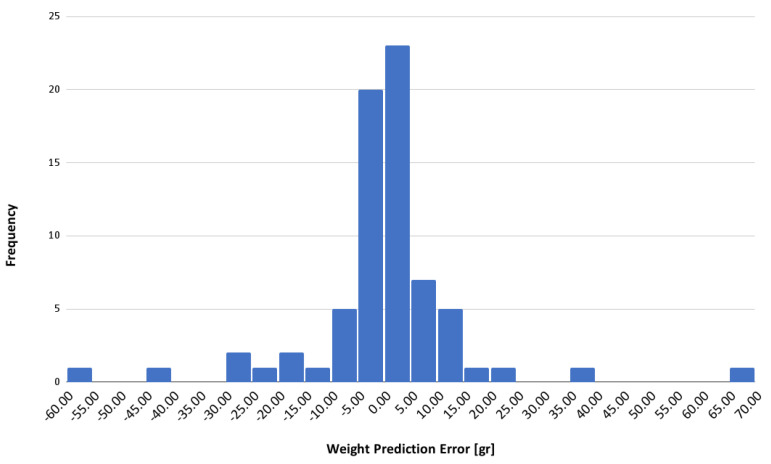
Signed error of rotisserie chicken weight prediction.

**Figure 14 sensors-24-07660-f014:**
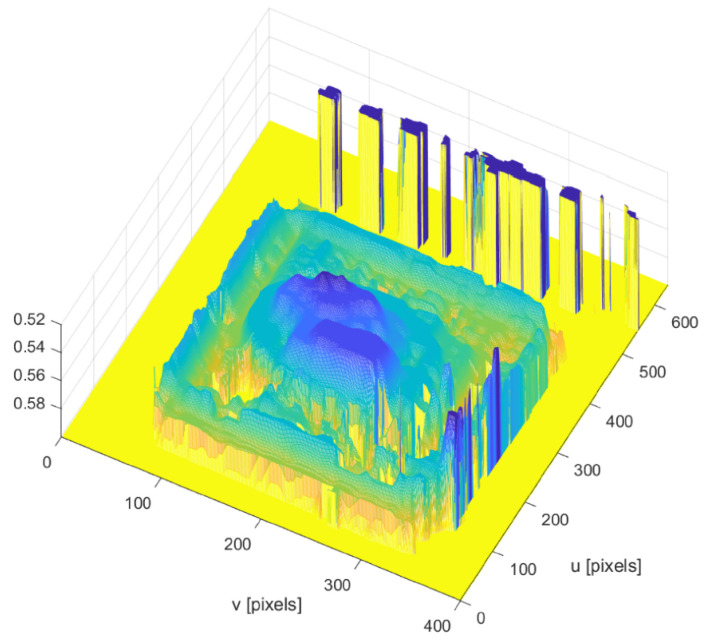
Depth image of configuration one of the full plates, including the outliers. The colors represent the depth values of the plate, where yellow indicates areas of higher depth values and blue represents lower depth values. Intermediate colors such as green and cyan reflect gradual transitions between these depth levels. The vertical spikes correspond to outliers in the data, which are represented as abrupt changes in depth.

**Figure 15 sensors-24-07660-f015:**
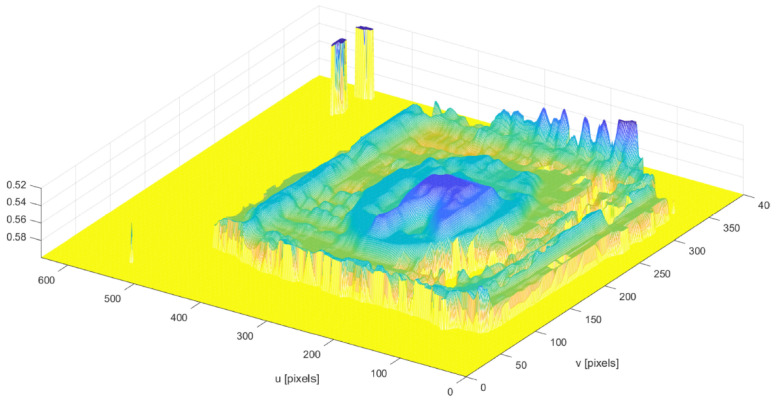
Depth image of correctly detected protein. The colors represent the depth values of the detected protein, where yellow indicates areas with higher depth values and blue represents lower depth values. Intermediate colors, such as green and cyan, reflect gradual transitions between these depth levels.

**Figure 16 sensors-24-07660-f016:**
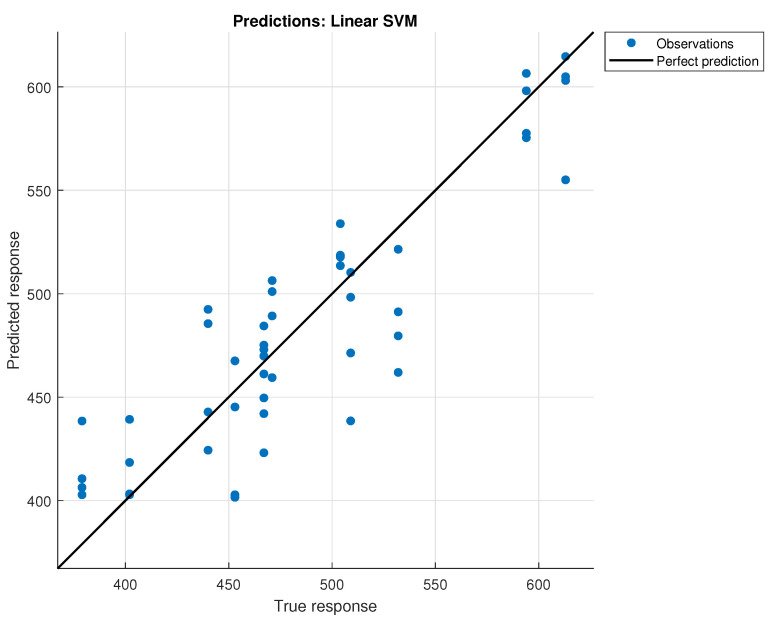
Predictions of the model for rice weight.

**Table 1 sensors-24-07660-t001:** Configuration of the OAK-D Lite depth camera used in the experiment.

Parameter	Value	Description
Resolution	1280 × 720 pixels	Resolution set for depth image capture.
Median filter	7 × 7	Filter applied to reduce noise in the image.
Temporal filter	alpha = 0.1	Temporal stabilization of depth measurements.
Spatial filter	alpha = 0.1, radius = 2, delta = 0	Edge smoothing with four iterations.
Confidence threshold	200	Confidence value to filter unreliable measurements
Decimation	Factor 2	Half of the images are considered to speed up processing.

**Table 2 sensors-24-07660-t002:** Models with Lower RMSE.

Model	RMSE
Exponential GPR	15.230
Rational Quadratic GPR	15.263
Squared Exponential GPR	15.492
Wide Neural Network	15.573
Linear SVM	29.803

**Table 3 sensors-24-07660-t003:** Comparative Analysis of Chicken Leg Volumes [mm^3^].

Config.	C1	C2	C3	C4	Std. Dev.
Comp.	254,145	239,256	277,954	286,303	21,609
1/2 Acc.	304,247	305,351	272,516	277,740	17,269
1/4 Acc.	262,466	286,887	279,901	274,849	10,299
1/8 Acc.	295,410	310,579	291,054	304,708	8840

**Table 4 sensors-24-07660-t004:** Models evaluated for the estimation of rice weight.

Model	RMSE (g)
Linear SVM	30.809
Exponential GPR	32.263
Rational Quadratic GPR	31.081
Squared Exponential GPR	31.115
Wide Neural Network	43.798

## Data Availability

Data is unavailable due to privacy restrictions.

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
