# Peer review of "Automated Food Weight and Content Estimation Using Computer Vision and AI Algorithms"

_sensors, 2024, doi:10.3390/s24237660_

Round 1

Reviewer 1 Report

Comments and Suggestions for Authors

It's unclear why the manual segmentation process is described in section 2.10. The paper is about automated food weight estimation. Manual segmentation can improve the results (similar to manual object detection and manual weighing of food). Still, as the paper proposes the automatic process, manual food segmentation should only be discussed as a method for training dataset preparation. 

The description of the Algorithm 1 is not clear.

Experimental evaluation is incomplete. There's no experimental evaluation of the pixel-level semantic segmentation component.

It's unclear how the ground-truth weights for evaluating the weight estimation model were obtained, and there is no estimation of how accurate this ground truth is. 

There are statements in the paper that are not supported by the evidence.  The authors should use simple and concise language and avoid overstating facts.

The authors write, "The volumetric data obtained from the previous steps serve as distinguishing features for training the predictive model. The model is designed to learn the complex relationship between the volume of the food item and its corresponding weight. This relationship is often non-linear, requiring a sophisticated modelling approach to capture the subtleties inherent in the data." - the model should estimate the weight of the food based on its estimated volume. Theoretically, it should be a linear relationship. To prove that, in practice, this relationship is indeed very complex and not easy to model; the authors should plot it for a few chosen food types - this would prove that it is indeed complex and cannot be modelled with simple models. Also, authors should evaluate simpler models such as linear regression.

It's unclear if the authors use a single model for estimating the weight from the volume or a different model for each food type.

The authors write, "This kernel formulation enables the model to effectively capture dependencies and continuity in the data, making it particularly suitable for our context where minor variations in texture and volume can significantly affect the weight estimation." It's unclear how the model can take variations in texture into account. I understand that the input to the model is the volume only, and the model is unaware of the texture.

What are the measurement units in Table 11?

Reference to the Kolmogorov-Arnold Network (KAN) is not necessary. It's not clear why KAN, in particular, have the potential to improve weight estimation techniques in the food industry. Same with LLM. It's not clear how LLMs can improve the food weight estimation form visual data. If authors know how these methods can improve the weight estimation from visual data they should clearly write it. Can these methods help with object detection? Semantic segmentation? Depth estimation? 

Comments on the Quality of English Language

The authors should use simple and more concise language.

Author Response

Thank you very much for your time and your comments. Please find enclosed our response

Reviewer 2 Report

Comments and Suggestions for Authors

The author provide a food weight estimation based on computer vision and deep learning, which is interesting. There are some area could be improved.

- The author lacks a lot of information about training (detection, segmentation), including platform, Labeling software, environment, software used, rounds, etc.

- The author mentions that the training set works well, but where are the results of the test set (pictures, tables)?

- It seems that the results only woke up the analysis of the chicken leg, where the rice?

- The author needs to give a diagram of where the RGB and depth cameras are installed. Are they together? Is there an image matching algorithm for depth images and RGB images?

- How much does the missing edge of the depth camera affect the weight estimate?

- How do you estimate the actual weight of the food when it blocks each other?

- If you rotate the plate, or rotate the same food, how much does it affect the weight estimate, because it doesn't seem to expand the data set.

Author Response

(The authors gave the same response as above.)

Round 2

Reviewer 2 Report

Comments and Suggestions for Authors

No more questions.